# Application of Thymol Vapors to Control Postharvest Decay Caused by *Penicillium digitatum* and *Lasiodiplodia theobromae* in Grapefruit

**DOI:** 10.3390/foods12193637

**Published:** 2023-09-30

**Authors:** Gabriela M. Olmedo, Jiuxu Zhang, Wei Zhao, Matthew Mattia, Erin N. Rosskopf, Mark Ritenour, Anne Plotto, Jinhe Bai

**Affiliations:** 1Horticultural Research Laboratory (USDA-ARS), 2001 S. Rock Rd, Ft. Pierce, FL 34945, USA; gabriela.olmedo@usda.gov (G.M.O.); wei.zhao@usda.gov (W.Z.); matthew.mattia@usda.gov (M.M.); erin.rosskopf@usda.gov (E.N.R.); anne.plotto@ars.usda.gov (A.P.); 2Indian River Research and Education Center, University of Florida, 2199 S. Rock Rd, Ft. Pierce, FL 34945, USA; jiuxuzhang@ufl.edu (J.Z.); ritenour@ufl.edu (M.R.)

**Keywords:** essential oil vapors, citrus, stem-end rot, green mold, sporulation

## Abstract

Two of the major postharvest diseases impacting grapefruit shelf life and marketability in the state of Florida (USA) are stem-end rot (SER) caused by *Lasiodiplodia theobromae* and green mold (GM) caused by *Penicillium digitatum*. Here, we investigated the in vitro and in vivo efficacy of vapors of thymol, a natural compound found in the essential oil of various plants and the primary constituent of thyme (*Thymus vulgaris*) oil, as a potential solution for the management of GM and SER. Thymol vapors at concentrations lower than 10 mg L^−1^ significantly inhibited the mycelial growth of both pathogens, causing severe ultrastructural damage to *P. digitatum* conidia. In in vivo trials, the incidence and lesion area of GM and SER on inoculated grapefruit were significantly reduced after a 5 d exposure to 50 mg L^−1^ thymol vapors. In addition, the in vitro and in vivo sporulation of *P. digitatum* was suppressed by thymol. When applied in its vapor phase, thymol had no negative effect on the fruit, neither introducing perceivable off-flavor nor causing additional weight loss. Our findings support the pursuit of further studies on the use of thymol, recognized as safe for human health and the environment, as a promising strategy for grapefruit postharvest disease management.

## 1. Introduction

Grapefruit (*Citrus paradisi*) is a citrus fruit grown in tropical and subtropical countries around the globe and is highly valued for its sour-to-semisweet flavor and its anticarcinogenic, anti-inflammatory, and antioxidative properties [1,2,3]. One of the main problems affecting citrus for fresh consumption and juice production is the significant economic loss caused by fungal pathogens that infect fruit before, during, or after harvest, which lead to disease at the postharvest stage. In the state of Florida (USA), hot and humid weather conditions favor the development of several postharvest diseases, but Diplodia stem-end rot (SER), caused by *Lasiodiplodia theobromae* (Pat.) Griffon & Maubl., and green mold (GM), caused by *Penicillium digitatum* Sacc., have always been the most prevalent [4,5]. Moreover, since Huanglongbing (HLB) was confirmed in Florida in 2005, citrus production in Florida has declined significantly largely due to the HLB epidemic [6], while a significant increase in the preharvest *Lasiodiplodia* colonization of HLB-affected fruit tissues has also been reported [7,8,9]. HLB, attributed to the bacterium *Candidatus* Liberibacter asiaticus, affects citrus trees, generating a multitude of symptoms that include twig dieback, leaf chlorosis and fruit size reduction, asymmetrical shape, irregular color, and premature drop. Typically, *L. theobromae* infects the citrus calyx and floral disk (button) during development on the tree but remains quiescent until the fruit is harvested. In HLB-infected trees, a more prolific *L. theobromae* presence was found in symptomatic fruit than in asymptomatic fruit, which may result not only in a higher postharvest SER incidence but also in exacerbated HLB-associated fruit drop in the field. A transcriptomic analysis revealed a gene expression profile in the abscission zone of HLB fruit that reflects typical characteristics of defense responses against necrotrophic fungal infection, which was validated by a phytohormone measurement and fungal detection [8]. Thus, the economic losses associated with SER appear to be aggravated in Florida since HLB spread throughout the state.

Traditionally, chemical fungicides have been the primary tool for the control of pre- and postharvest diseases. Postharvest fungicides thiabendazole, imazalil, and fludioxonil are effective in reducing GM, and they have also shown some efficacy against the SER causal agent [4]. However, the year-round usage of the same active ingredients is highly conducive to the selection and proliferation of resistant biotypes of fungal pathogens, such as *P. digitatum* [10]. Moreover, fungicide use is increasingly restricted due to the associated risks to human health and negative environmental impacts. Therefore, growers and packers are left with few effective and environmentally accepted fungicides, and there is an urgent need to search for other safe and effective alternatives that also can reduce the development of pathogen resistance [11,12]. One such alternative is the exploitation of essential oils (EOs), which are known to have little to no impact on human health or the environment. These plant aromatic compounds are generally recognized as safe (GRAS), and research into their use in postharvest disease control has expanded in the past decade. The utilization of EOs and their main chemical components has resulted in the control of postharvest diseases on a variety of fresh fruits and vegetables [13,14,15,16], but many were reported to have perceivable negative sensory impacts and/or induced phytotoxic damage to the peel [17,18,19]. While much previous work was conducted by incorporating EOs into wax coatings or dipping fruit into EO solutions, the application of EOs in the vapor phase has been shown to have certain advantages over that in the liquid phase, such as a minimal or no negative impact on products’ organoleptic properties, improved penetration, rapid and homogeneous distribution, and enhanced antimicrobial activity [20,21,22,23]. Here, we evaluated the effect of thymol, the main constituent of thyme (*Thymus vulgaris*), in its vapor phase on *P. digitatum* and *L. theobromae*. For this purpose, pathogen mycelia and conidia were exposed to thymol vapors, and inhibitory concentrations were calculated. The in vivo efficacy of thymol treatments to control SER and GM was tested on grapefruit (*Citrus paradisi*) inoculated with fungal pathogens. After storage, weight loss and sensory perceptions were assessed.

## 2. Materials and Methods

### 2.1. Fungal Pathogens and Fruit

The isolates used in this study were obtained from the Department of Horticultural Sciences at the University of Florida, India River Research and Education Center, in Fort Pierce, FL (USA). *P. digitatum* (isolate PD-26) was originally collected by G.E. Brown (Florida Department of Citrus, Lake Alfred, FL, USA) and is resistant to thiabendazole [24]. *L. theobromae* (isolate D-36) was isolated from red grapefruit (*C. paradisi*) showing typical lesions of Diplodia stem-end rot (SER) from a grove located in Fort Pierce, FL. For both isolates, morphological identification and pathogenicity tests following Koch postulates on grapefruit were performed [25]. Isolate identities were confirmed via molecular methods using ITS4/ITS5 and EF1-728F/EF1-986R primer sets to amplify the internal transcribed spacer (ITS) region in both isolates and part of the translation elongation factor 1-alpha (TEF1-α) gene in *L. theobromae*. Briefly, mycelia grown on potato dextrose broth were homogenized under liquid nitrogen, and the DNA was extracted using a DNA easy Plant Mini Kit according to the manufacturer’s protocol (Qiagen Inc., Germantown, MD, USA). A total of 50 µL PCR mixture was prepared for each isolate, containing 25 µL 2× GoTaq Green Master Mix (Promega, Madison, WI, USA), 2 µL forward and reverse primers (10 µM), 19 µL nuclease-free water, and 2 µL genomic DNA (25–65 ng mL^−1^). PCR reactions were run in a T100 Thermal Cycler (Bio-Rad, Hercules, CA, USA) that was programmed for a 5 min initial denaturation step at 94 °C, followed by 40 cycles (each 45 s at 94 °C, 30 s at 52 °C, and 90 s at 72 °C) and a 6 min final extension at 72 °C. The amplified products were purified using a QIAquick PCR Purification Kit (Qiagen Inc., Germantown, MD, USA) and sequenced using a 3730xl DNA Analyzer (Applied Biosystems, Hercules, CA, USA). The obtained sequences were compared to the ones reported for *L. theobromae* and *P. digitatum* in GenBank.

The grapefruits (*C. paradisi*, var. ‘Flame’) used for the in vivo experiments were harvested from commercial groves in Saint Lucie County (FL, USA). Fruits with intact stems and no wounds were selected, taken to the US Horticultural Research Laboratory (USDA-ARS), superficially sanitized via immersion in 200 mg L^−1^ NaClO (pH 6) for 2 min, and then air-dried before use.

### 2.2. Chemical Compounds

Thymol (98.5%) was purchased from Sigma-Aldrich Co. (St. Louis, MO, USA) and kept at room temperature in the dark until use. For every assay, a thymol melting/crystallization cycle was performed by heating thymol powder above its melting point (49.6 °C) following a protocol reported by Cometa et al. [26] with slight modifications. Namely, thymol was subjected to a thermal treatment at 55 °C until melted onto a 20 mm diameter filter paper (No. 4, Whatman, Cleves, OH, USA). The filter paper was immediately attached to the inner surface of the top lid of the container that was used as indicated for each assay (Petri dishes or plastic 11 gal boxes). The containers were immediately closed and sealed.

### 2.3. Effective Inhibitory Concentration Determination

In vitro fungal growth inhibition using thymol was assayed following the same protocol for both pathogens, but the initial fungal structures differed (Figure 1). For *L. theobromae*, a 3 mm diameter mycelial plug was placed in the center of a Petri dish (9 cm diameter and 5 cm in height) containing potato dextrose agar (PDA), while for *P. digitatum*, a 5 μL spot of conidial suspension containing 10^4^ conidia mL^−1^ was used for plate inoculation. The difference in methodology is based on each pathogen’s disease cycle. The key role of conidia in *P. digitatum* spread and infection has been previously reported [27]. Asexual spores are the primary inoculum for GM during fruit–pathogen interaction since the germination of conidia occurs on the surface wounds of citrus. Conversely, *L. theobromae* development on fruit begins in the field, but mycelia remain quiescent until harvest. When the senescence of stem-end tissue post-abscission proceeds, which may be enhanced by postharvest ethylene degreening treatments and under conducive environmental conditions (such as high temperatures and relative humidity), fungal hyphae resume growth, colonize and infect fruit tissue adjacent to the stem-end, and develop through the fruit core tissues, causing whole fruit decay [4]. After inoculation, thymol treatments were applied by melting and crystalizing thymol as explained previously. To achieve the final concentrations of 1, 2, 4, 8, and 16 mg L^−1^ within plate headspaces, 0.32, 0.64, 1.28, 2.56, and 5.12 mg of thymol were used, respectively. The plates were sealed with parafilm and incubated at 25 °C in the dark. Colony diameters were measured daily until the mycelia in the control plates without thymol covered the entire plate. Thymol concentrations that inhibited 50 and 90% of the mycelial growth (effective concentrations, EC_50_ and EC_90_) were calculated based on logarithmic models (based on the relationship between percent inhibition and compound concentration) [28].

### 2.4. Scanning Electronic Microscopy (SEM) Observation

SEM was used to observe *P. digitatum* conidial morphology and cell wall integrity after treatment with thymol at sublethal concentrations. Conidial suspensions were exposed to 1 mg L^−1^ thymol treatment for 24 h and then observed under a microscope. Sample preparation was performed following a previously reported protocol [29], with some modifications. Cells were washed three times with deionized water and then covered with a fixative solution containing 2.5% glutaraldehyde and 4% paraformaldehyde in sodium cacodylate buffer 0.1 M (pH 7) (Sigma Aldrich Co., St. Louis, MO, USA). Tubes were incubated at 4 °C overnight. Then, the cells were again washed three times and dehydrated by consecutively transferring them to increasing ethanol concentrations from 30% to 100%. Incubation with 100% ethanol was carried out three times. Finally, the samples were transferred to solutions of ethanol:hexamethyldisilizane (HMDS, Sigma Aldrich Co., St. Louis, MO, USA) at ratios of 3:1, 2:1, and 1:1 with two final steps at 100% HMDS and allowed to evaporate in a chemical fume hood overnight. The cells were attached to SEM stubs, and observations were performed using an S-4800 scanning electron microscope (Hitachi High Technologies America, Inc., Pleasanton, CA, USA) with 1000× magnification and a 5 kV accelerating voltage.

### 2.5. In Vivo Assays in Airtight Conditions

For in vivo assays, the fruits were inoculated with the pathogens, as shown in Figure 1. For *L. theobromae*, the button on the stem-end of each fruit was removed to expose fresh tissue. A 3 mm diameter mycelial PDA plug from the margin of a 3 d old fungal colony was placed with mycelia facing down on the wounded area. After 6 h at 25 °C, the plugs were removed. For *P. digitatum*, the fruits were inoculated using a 1 mm wide and 2 mm long sterile stainless steel rod previously immersed in a freshly prepared conidial suspension with a concentration of 10^4^ conidia mL^−1^ [30]. After inoculation, the fruits were placed inside 11-gallon gasket-sealed storage boxes (Weather Shield Storage Box, Ziploc, S. C. Johnson & Son, Racine, WI, USA), with twelve fruits per container, arranged in a single layer. Thymol-containing filter papers were attached to the lids, as explained above, to achieve concentrations ranging from 1 to 100 mg L^−1^. This concentration range was based on antifungal in vitro results, considering that antimicrobial in vivo efficacy is often lower than antimicrobial in vitro efficacy. A thymol slow-release treatment was included to determine whether a more extended inhibitory effect was achieved. This treatment consisted of a Miracloth (Calbiochem^®^, San Diego, CA, USA) sachet containing pectin–alginate microencapsulated thymol prepared following a protocol previously reported by our group [31]. A 40 mm USB cooling fan (5 V, 5300 rpm, Shenzhen Engesen Electronics Co., Ltd., Shenzhen, China) was attached to an inner wall of each container to ensure air circulation. The containers were closed and stored at 25 °C and 90% RH for 5 d prior to disease incidence and lesion area evaluation. The filter papers were then removed, and the fruits were maintained in the open air for an additional 5 d at 25 °C to determine whether the thymol vapor had a fungicidal or fungistatic effect. Controls consisted of inoculated fruit stored in the same conditions as treatments but not exposed to thymol.

### 2.6. Weight Loss and Sensory Evaluation

As a quality parameter of grapefruit after thymol treatment, individual fruit weights were determined before and after 5 d of storage at 25 °C. Weight loss was calculated as (A_0_ – A_x_)/A_0_) × 100, where A_0_ is the initial fruit weight, and A_x_ is the weight at the corresponding evaluation time.

All treatments were first evaluated for the potential effects of the carryover of thymol on fruit flavor, and three treatments were chosen for a sensory evaluation test: (i) untreated control, (ii) 100 mg L^−1^ thymol, and (iii) 100 mg L^−1^ thymol encapsulated in pectin–alginate. Following storage, 10 grapefruits per treatment were washed, sanitized in 0.01% PAA, and air-dried at room temperature overnight. On the following day, the fruits were peeled, removing the stem and blossom ends, flavedo, and albedo, and cut into 14–16 pieces each. The pieces were first mixed in a fruit bowl to ensure that every panelist would obtain a 3-piece sample representing different fruits from each treatment. The samples were placed in 3-digit-coded 4 oz cups on a serving tray and served at 18 °C in a randomized order following a Williams design, with the order of presentation balanced across panelists. Testing took place in isolated booths equipped with positive air pressure and red lighting. Sensory evaluation was carried out by a panel of 24 untrained staff members, some of whom were familiar with tasting various citrus fruits. Every panel member was asked to rank the samples for overall preference (“like most” to “like least”) [32]. In a second question, they were asked, for each sample, to indicate whether they could detect any off-flavor and describe that off-flavor if any was perceived.

### 2.7. Statistical Analysis

For in vitro assays, three Petri dish replicates were performed for each condition, and the entire panel of assays was performed three times. In vivo experiments were conducted with a completely randomized design, with two replicates consisting of 12 fruits per replicate (container with 12 grapefruits). The experiment was performed twice. In all cases, an analysis of variance was used to check homogeneity and normality, followed by Student’s *t* least significant difference (LSD), and mean separations were based on *p* values ≤ 0.05. Sensory rank data were analyzed using the Friedman-type statistical test for rank data, with the non-parametric analog to Fisher’s LSD for rank sums [32].

## 3. Results

### 3.1. Penicillium digitatum Conidia Germination and Lasiodiplodia theobromae Mycelial Growth Inhibition by Thymol Vapors In Vitro

The antifungal activity of EOs has become an important area of agricultural research due to the increasing awareness of the human and environmental toxicity associated with traditional fungicides, as well as the increasing appearance of multi-resistant fungal isolates in packinghouses throughout the world [33,34]. In this work, we evaluated the effect of thymol vapors on two major fungal pathogens affecting citrus in Florida (USA). As shown in Figure 2, fungal growth was significantly affected by exposure to thymol in a concentration-dependent manner. While both pathogens were completely inhibited by 8 mg L^−1^ of thymol, effective concentration calculations showed that *P. digitatum* (EC_50_ = 0.135 mg L^−1^ and EC_90_ = 1.235 mg L^−1^) was more sensitive than *L. theobromae* (EC_50_ and EC_90_ of 1.13 and 5.42 mg L^−1^, respectively).

It is noteworthy that, while *P. digitatum* mycelial growth was not significantly inhibited by 1 mg L^−1^ thymol, sporulation was severely affected, leading to a plain white colony (Figure 3a), compared to the characteristic full green color associated with conidia in the control plates (Figure 3b), although this inhibition was not quantified. An SEM observation showed that EO at 1 mg L^−1^ (sublethal concentration) caused severe ultrastructural damage to *P. digitatum* conidia during the first 24 h. The conidia in control samples exhibited a regular and homogenous morphology consisting of smooth and continuous cell walls (Figure 3c), while after exposure to EO vapors, most cells were shrunken and abnormally shaped (Figure 3d). Also, considerable quantities of cellular debris were observed in the thymol-treated samples, as an indication of conidial lysis. As mentioned previously, *L. theobromae* growth was delayed by the thymol vapors (Figure 2); however, colony characteristics remained the same as those observed under control conditions.

The effective concentrations found in the present work were significantly lower than those reported by other authors. For instance, Ding et al. [35] found that EC_50_ values ranged between 29.8 and 55.33 mg L^−1^ for six different postharvest disease-causing fungi, including *Penicillium* sp. Also, Zhang et al. [36] reported a minimum inhibitory concentration of 65 mg L^−1^ against *B. cinerea*, and Yan et al. [37] reported an EC_50_ of 37 mg L^−1^ against *L. theobromae*. The primary difference between previous studies and the present one, other than the pathogen strains used, is the thymol application method used; previous authors added thymol directly into a liquid medium or dissolved it in ethanol before incorporating it into the medium (diffusion methods). Here, pathogens were directly exposed to the EO vapors produced after heating at temperatures above thymol’s melting point. The results of the current study are in agreement with the results of various studies in the medical field in which EO vapors exhibited greater antimicrobial effects than EOs delivered in liquid form applied by direct contact [38,39,40,41]. In addition, Boukhatem et al. [42] reported that micellar formation by lipophilic molecules in the aqueous phase reduces EO accessibility to microorganisms, whereas the vapor phase allows free attachment. These observations might explain the lower EC50 and EC90 concentrations observed here.

### 3.2. Postharvest Disease Control by Thymol Vapor on Inoculated Grapefruit

The evaluation of the potential of thymol to control postharvest diseases in grapefruit was assessed in airtight containers at room temperature over a period of 5 d. When the fruits were inoculated with *L. theobromae* or *P. digitatum* and exposed to direct EO vapors, concentration-dependent incidence reductions were observed (Figure 4). The effective concentrations required to reduce in vivo pathogen growth were 10 to 50 times higher than those in vitro. It is commonly reported that in vivo efficacy is often reduced when compared to in vitro studies, particularly in the application of antimicrobial compounds and specifically for EO vapors applied for antimicrobial treatments in fresh food studies, including citrus [43] and cherry tomatoes [44], as well as in processed food products, like cheese [45], bread [46], mushrooms [47], and coffee beans [48]. As was observed in the in vitro assays, *P. digitatum* was more sensitive to thymol than *L. theobromae*. For instance, 50 mg L^−1^ thymol treatment resulted in an 81% reduction in SER incidence and a 90% reduction in visible lesion size (Figure 4, lower panels), while the same treatment on *P. digitatum*-inoculated fruit resulted in 90 and 99% reductions in GM incidence and lesion size, respectively (Figure 4, top panels). In contrast, the controlled-release treatment of thymol did not exhibit any inhibitory effect against the pathogens. This suggests that, after inoculation, fruit infection developed at a rate that did not allow the controlled-release thymol to prevent pathogen reproduction.

The different effects of thymol vapors on the assayed pathogens in this study could be explained by the different growth patterns in fruit and, thus, how much vapor can reach the mycelia. Edris and Farrag [49] stated that superficial-growing molds are particularly susceptible to EO volatile compounds. It is known that *L. theobromae* infects fruit from the button at the stem-end and proceeds through the core more quickly than the rind [4]. However, *P. digitatum* infection begins with a wound on the rind and spreads on a number of skin oil glands through pores, remaining relatively superficial during the first few days [50]. Based on these characteristics, thymol vapors might be in direct contact with *P. digitatum* cells more readily than *L. theobromae* cells growing underneath the rind through the core.

After the 5 d initial storage, the fruits were kept for five additional days in the open air and again evaluated (Figure 5). After 10 d, the GM incidence in the untreated control remained around 90%, and decayed fruits were fully covered by olive-green conidia. In Figure 5, the lower left panel shows that, after five additional days, nearly all grapefruits exposed to 10 mg L^−1^ thymol developed decay, which suggests that thymol’s effect is fungistatic rather than fungicidal. However, the pathogen growing on these fruits was not able to produce conidia, confirming the anti-sporulating effect observed in in vitro trials (Figure 3a,b). When developing in a single fruit, *P. digitatum* usually produces one to two billion greenish conidial spores that are dispersed into the atmosphere, perpetuating disease [10]. These conidia are particularly difficult to inactivate due to their stability when exposed to heat, light, and chemical reagents [51]. While further assays are needed to elucidate the exact mechanism involved in the sporulation impairment achieved by thymol, this result is worth highlighting, as it may be crucial for improved postharvest disease control.

The *P. digitatum* isolate used in this work is resistant to thiabendazole [24]. This is, as mentioned before, one of the major challenges faced by the agricultural industry worldwide [52]. It has been reported that the mode of action of thymol is broad and non-specific; i.e., it induces changes in several metabolic pathways affecting DNA synthesis, cytoplasmic and outer membrane permeability, and fatty acid and volatile compound profiles [53]. Hence, microbial resistance development against thymol vapor treatments is unlikely to occur, which further supports their potential use as alternatives and/or complements to current postharvest microbial management strategies. Among the various advantages of the vapor-phase applications of EOs, the gas improves penetration and homogeneity and can potentially provide a more consistent and uniform pathogen exposure to thymol, even in hard-to-reach areas. Also, vapor treatments do not require solvents or carriers that might be needed for solution-based treatments, and, thus, potential side effects or interactions between pathogens and those additional ingredients are reduced.

### 3.3. Fruit Weight Loss and Sensory Evaluation

The exposure of the grapefruit to different thymol concentrations did not alter the fruit weight loss when compared with the untreated fruit. The fruit weight loss was lower than 0.4% after 5 days at 25 °C in all cases. Also, no phytotoxicity was observed after the thymol vapor treatments. Previous research has shown that, when essential oil vapor is applied to fruits, the effects on weight loss can indeed be varied, depending on various factors, including the essential oil type, vapor concentration, fruit type, and environmental conditions. A weight loss reduction is most likely related to EOs’ antimicrobial properties, as well as their ability to form a protective layer on the surface of the fruits, which prevents moisture loss [54]. Contrarily, an increase in weight loss is mostly due to phytotoxicity caused by a high EO dosage, which damages the fruit skin, or altered respiration [17,55]. Our observations indicate that the applied thymol vapor dosages did not harm the fruit surface. However, the low dosage also did not effectively prevent water loss.

As mentioned before, one of the main limitations for the use of EOs in food processing is the negative impact that they might have on organoleptic properties. Thus, we evaluated the panelists’ sensory perception after grapefruit exposure to thymol vapors at 100 mg L^−1^ of thymol pure or microencapsulated in pectin–alginate. In the taste panels, the rank sums for preference were 51, 50, and 43 points for 100 mg L^−1^ of thymol, microencapsulated thymol, and the control, respectively, with no significant difference between treatments (T = 1.583, Friedman statistics for rank data). Furthermore, the panelists’ comments after tasting suggested that there were no perceived off-flavors that could be related to the thymol treatments. The lack of undesired flavors observed in the present work is likely due to the dosages of the thymol vapors being low enough; thus, accumulation on the fruit surface did not exceed the threshold levels. Also, the treatment was applied to grapefruit, which has a rather thick peel acting as a barrier to the edible part. It would be appropriate to test the treatment in thinner-peel fruit such as mandarins. Nevertheless, our results agree with those of previous reports where essential oil vapors were used to inhibit different bacterial and fungal strains that cause food spoilage, without exceeding the acceptable flavor thresholds [56,57,58]. Under our experimental conditions, the application of thymol in the vapor phase would not lead to consumer rejection.

## 4. Conclusions

Our findings demonstrate the significant inhibitory effects of thymol vapors on the two primary postharvest pathogens responsible for decay in commercial grapefruit. While thymol alone may not completely control SER and GM, its application resulted in noteworthy reductions in disease incidence and hindered the production of conidia by *P. digitatum.* Also, under our experimental conditions, no negative effects, such as off-flavors or a weight loss increase, resulted from the thymol vapor applications. These positive outcomes support the potential implementation of this EO as part of a grapefruit postharvest management strategy. Future studies involving combinations of thymol vapors and postharvest fungicides, as well as other postharvest fruit handling processes, will be conducted to analyze the feasibility of its application at commercial scale. Considering this compound’s known safety for human health and the environment and the several advantages of its application in the vapor phase compared to in liquid form, our work presents a promising option to complement current postharvest management approaches.

## Figures and Tables

**Figure 1 foods-12-03637-f001:**
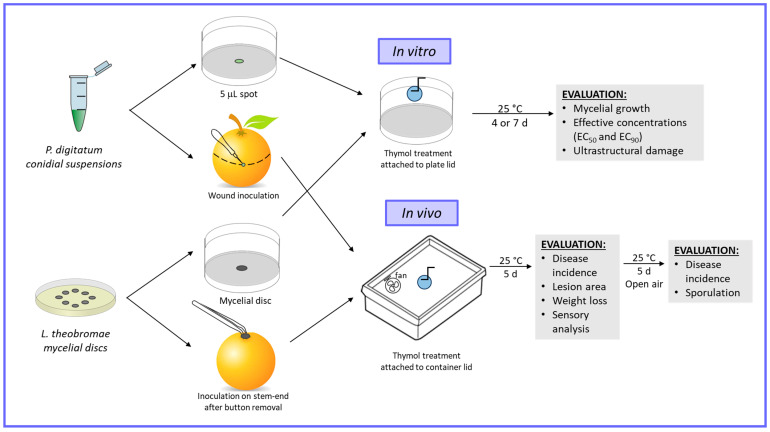
Protocols for the evaluation of the effect of thymol vapors on pathogens’ in vitro growth and in vivo development on grapefruit. *P. digitatum* conidial suspensions (10^4^ conidia mL^−1^) and *L. theobromae* mycelial discs (3 mm diameter) were used to inoculate PDA plates and grapefruit. Thymol treatments were applied at 25 °C in air-tight conditions. After incubation, pathogens’ in vitro growth and ultrastructural damage were evaluated. In in vivo experiments, disease incidence, lesion area, weight loss, and sensory analyses were performed after 5 d. Then, containers were opened, thymol discs were removed, and fruits were stored for 5 additional days to determine whether thymol vapor had fungicidal or fungistatic effects.

**Figure 2 foods-12-03637-f002:**
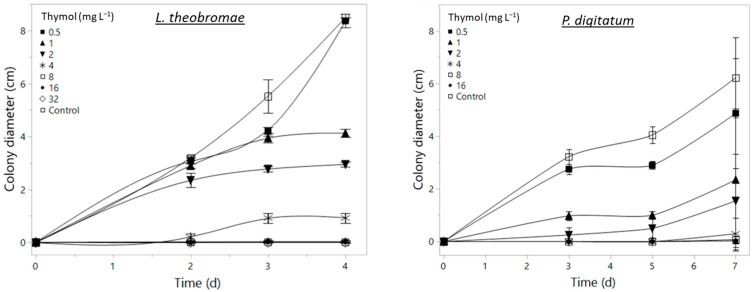
*Lasiodiplodia theobromae* and *Penicillium digitatum* growth inhibition resulting from increasing thymol vapor concentrations in sealed Petri dishes on potato dextrose agar. Plates were stored at 23 °C, and colony diameter was measured daily.

**Figure 3 foods-12-03637-f003:**
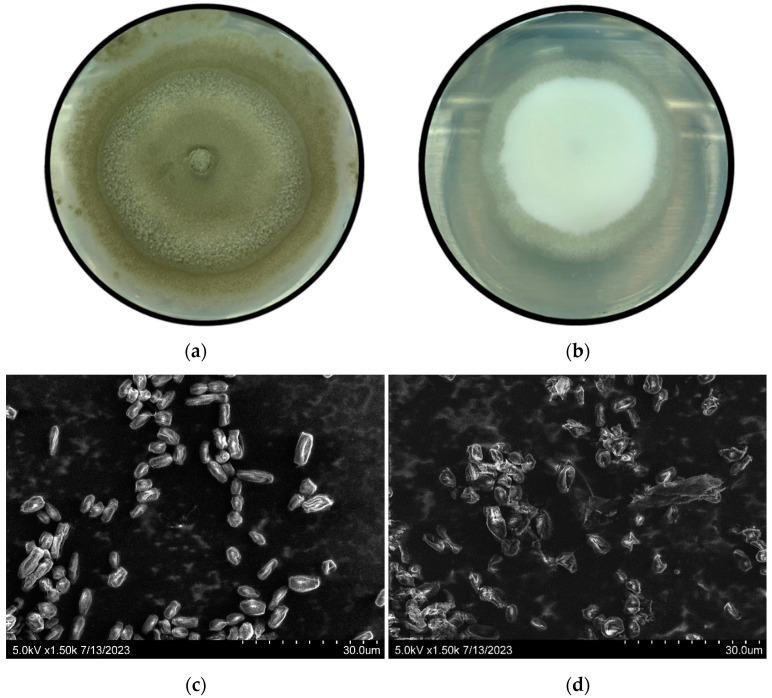
Effect of thymol vapor on *Penicillium digitatum* mycelia and conidia. Suspensions of 10^4^ conidia mL^−1^ were exposed to thymol vapor at 23 °C. Top and lower panels represent colony morphology after 4 d and conidia ultrastructural characteristics (SEM photographs at 1500× magnification) after 24 h, respectively, for control condition (**a**,**c**) and 1 mg L^−1^ thymol treatment (**b**,**d**).

**Figure 4 foods-12-03637-f004:**
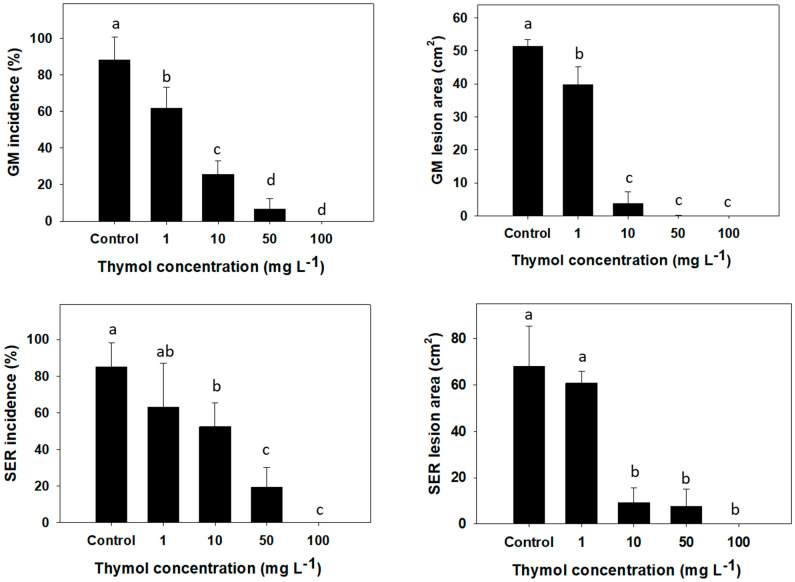
Incidence and lesion area of green mold (GM) and stem-end rot (SER) on grapefruit inoculated with *Penicillium digitatum* (GM) and *Lasiodiplodia theobromae* (SER), exposed to thymol vapors at 1, 10, 50, and 100 mg L^−^^1^ for 5 d at 23 °C. Top panels: GM; bottom panels: SER. Columns with different letters are significantly different according to Student’s *t* LSD (*p* ≤ 0.05).

**Figure 5 foods-12-03637-f005:**
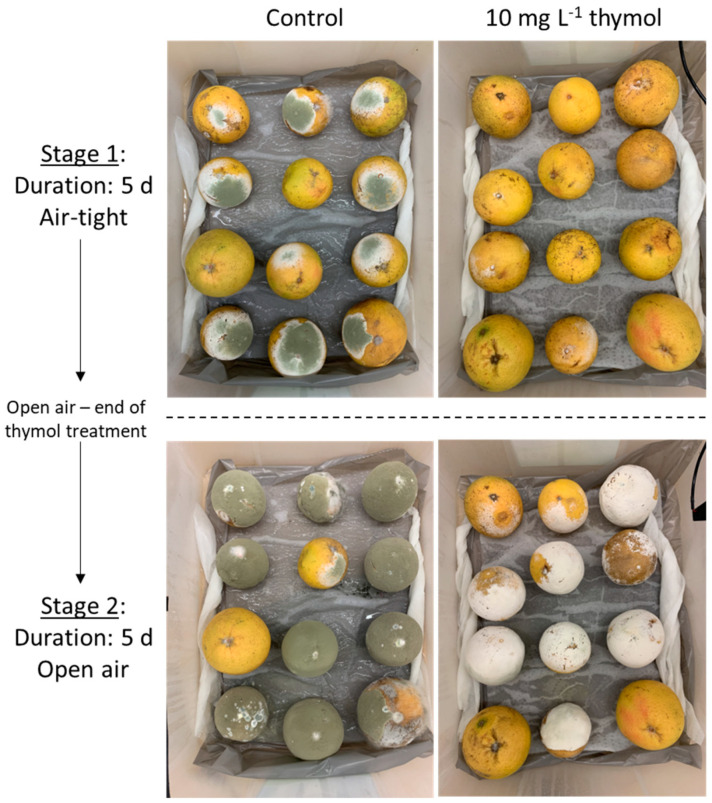
*Penicillium digitatum*-inoculated grapefruits exposed to 10 mg L^−1^ thymol. Pictures show control (**left** panels) and 10 mg L^−1^ thymol treatment (**right** panels) after 5 initial days in air-tight conditions (**top**) and 5 additional days in the open air (**bottom**) at 23 °C.

## Data Availability

The data presented in this study are included in the article. Further inquiries can be directed to the corresponding author.

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
