# Peer review of "Application of Thymol Vapors to Control Postharvest Decay Caused by Penicillium digitatum and Lasiodiplodia theobromae in Grapefruit"

_foods, 2023, doi:10.3390/foods12193637_

Round 1

Reviewer 1 Report

The manuscript is not suitable to be reviewed in this state. It must be reorganized in the foods format. There  are several technical issues in the formatting. It is not mentioned what kind of article is this?. The abstract looks like a passage which lacks the numerical data. How much Thymol controlled the spoilage of tested fruit?.. so on. Please revise the content in foods format. and resubmit after major revision.  

Minor English changes required. 

Author Response

Response: We carefully addressed your and other reviewers’ comments, made corrections, and improved the overall readability and quality of the manuscript. We also fitted the manuscript to the Foods template. As stated in the new version, the manuscript type is “Article”. We checked that critical numerical data are present in the abstract, such as thymol concentrations that significantly inhibited in vitro growth of both, L. theobromae and P. digitatum, as well as which thymol treatments were able to reduce green mold and stem-end rot development on inoculated grapefruit. Also, effects on P. digitatum suppression, sensory and weight loss evaluations on tested fruit are mentioned.

Reviewer 2 Report

Dear editor and authors,

The authors of the article entitled: Use of thymol vapors to control Diplodia stem-end rot and green mould on grapefruit after harvests dealt with testing the gas phase of thymol as a potential inhibitor of the microscopic filamentous fungi L. theobromae and P. digitatum in the  in vitro and in vivo. The article is clearly written, and we can see that the authors have done a lot of work, but it also has its shortcomings. The first is that the authors didn't use a template (it wouldn't matter too much, it can be done in the review process), but I recommend putting the article in the Foods template. I have a few comments on the article, which should be explained and important information added to the manuscript and the material and methods section before further processing the article:

·     -First of all, I miss novelty of this article! I recommend that authors include it at the end of the introduction!

·        - Use the correct Latin names of the plants (C. sinensis L.)

·        - what concentration was used in the in vivo experiments? I can't find it in the methodology, please fill it in and explain on what basis you chose such a concentration!

·        - How long did it take the thymol to evaporate? Have you done any optimization? On what basis can you say that the vapors of thymol remained over the fruit or over the mushroom mycelium for 5 days?

·         -According to which standards and where was the sensory analysis carried out? Did you use the triangular method? Add this information to the Materials and methods section.

·       -  Why is L. theobromae tested for 4 days and P. digitatum for 7 days (in Figure 1)?, I don't see an explanation anywhere! See the reasons for the different culture and culture method in the Materials and methods section.

·       -  The names of all the pictures have been moved.

·         -Why did you observe the effect of thymol vapor only on the mycelium and conidia of P. digitatum?? Explain in the Materials and methods section and also in the Results and discussion!

·        - Where are the results of the sensory evaluation? I did not find any in the article. The authors only state that the vapors had no effect on the sensory system, but we do not know that.

·        - Add more recent literature to the article, in the journal FOODS there are several publications devoted to the gas phase of essential oils and their influence on the mycobiota, for example coffee, cheese, etc...     

·        - I recommend adding a pictorial appendix for both in vivo and in vitro experiments.

·        - I would also recommend adding sensory analysis and study limitations and future perspectives to the conclusion!

Overall, I rate the article positively, but it has major shortcomings in the description of section Materials and methods and missing novelties! I recommend the authors to include all the notes to improve their manuscript.

Moderate editing of English language is required.

Reviewer 3 Report

In this manuscript, the effects of thymol vapors against Lasiodiplodia and Penicillium were investigated. It is valuable in industry to some extent. However, some points should be clarified. The details as below,

1. The title can be more accurate.

2. Abstract should be more concise because it is very important to attract the readers.  Keywords should be added based on the contents.

3. line 31, the citation is not correct. What is the 80-90% representing in line 33.

4. The authors mentioned that the essential oil side effects on application, and was usually used as coating or dipping. What is the consideration using the vapor?

5. The authors monitored the weight loss and sensory evaluation, and they find the application has neither off-flavor nor weight loss, why?

6. How to obtain the treated concentration?

7. In vitro, what is the function for effective inhibitory concentration determination?

8. line 159, fruit should be "fruits", double-check through the manuscript.

9. Figure 3, GM lesion area, the significance label should be checked between 10 and 50 mg/L concentrations.

10. line 233-235, it should be moved up.

11. The writing should be checked carefully, and the quality should be improved. 

Some spelling mistakes should corrected.

Author Response

In this manuscript, the effects of thymol vapors against Lasiodiplodia and Penicillium were investigated. It is valuable in industry to some extent. However, some points should be clarified. The details as below,

  1. The title can be more accurate.

Authors to R#1: Thank you for your interest and careful review of our manuscript. As suggested, title was changed to Application of Thymol Vapors to Control Postharvest Decay Caused by Penicillium digitatum and Lasiodiplodia theobromae in Grapefruit..

  1. Abstract should be more concise because it is very important to attract the readers. Keywords should be added based on the contents.

Authors to R#2: Thank you for this suggestion. Abstract was improved and keywords were corrected to engage readers.

  1. line 31, the citation is not correct. What is the 80-90% representing in line 33.

Authors to R#3: The idea in this sentence was rewritten to make it clearer. Since HLB was confirmed in Florida in 2005, the industry has lost 80-90% of its production. It has been reported that L. theobromae pre-harvest colonization on HLB symptomatic fruit tissues is higher than in asymptomatic tissues. Therefore, since HLB has spread in Florida, SER incidences and premature fruit drop have been aggravated.

  1. The authors mentioned that the essential oil side effects on application and was usually used as coating or dipping. What is the consideration using the vapor?

Authors to R#4: As requested by you and R#2, the benefits of applying thymol in the vapor phase in comparison with coating and dipping applications are described in lines 68-72 of the introduction, as well as in the Results and Discussion, lines 263-266 and lines 349-354.

  1. The authors monitored the weight loss and sensory evaluation, and they find the application has neither off-flavor nor weight loss, why?

Authors to R#5: Thank you for this question. We decided to monitor these two postharvest features (standard postharvest quality evaluations) to make sure that thymol in vapor phase will reduce decay incidences as well as keep or not negatively affect marketability and consumers acceptance.

In the sensory evaluation, we have selected the dosage that controls disease without causing off-flavor, namely, the effective antifungal dosage that is lower than the human being detectable threshold.

On the other hand, we monitored weight loss to make sure that thymol vapor treatment reduced fruit decay without aggravating or abbreviating weight loss in comparison to the control.

  1. How to obtain the treated concentration?

Authors to R#6: Our treatments concentrations were calculated considering initial thymol amounts and containers volume.

  1. In vitro, what is the function for effective inhibitory concentration determination?

Authors to R#7: Effective thymol concentrations (EC50 and EC90) were calculated using a probit regression procedure, adjusting linear regression equations with thymol concentrations versus growth inhibition percentages.

  1. line 159, fruit should be "fruits", double-check through the manuscript.

Authors to R#8: This correction was applied (line 221 in revised manuscript) and checked through the manuscript.

  1. Figure 3, GM lesion area, the significance label should be checked between 10 and 50 mg/L concentrations.

Authors to R#9: Thank you. We have carefully checked the significance labels in all graphs.

  1. line 233-235, it should be moved up.

Authors to R#10: We moved the figure title next to the figure.

  1. The writing should be checked carefully, and the quality should be improved.

Authors to R#11: Thank you for your useful corrections. English was improved, and typing mistakes were corrected.

Round 2

Reviewer 3 Report

  1. The authors have responded to the reviews' comments point to point. So it read better and clearer. However, about the question 5, it is not enough strong yet. The authors monitored the weight loss and sensory evaluation, can they explain what is the mechanism or deep reason to cause their positive effect (neither off-flavor nor weight loss). They don't need to repeat the results, we believe the results too, just explain what is the possible reason to the perfect application without any side-effects?

Author Response

Thank you very much for your suggestions. The possible reasons for the lack of weight loss and undesired sensory effects were further discussed (see Lines 361-369 and 378-384).